# Onychomycosis in Foot and Toe Malformations

**DOI:** 10.3390/jof10060399

**Published:** 2024-05-31

**Authors:** Eckart Haneke

**Affiliations:** 1Schlippehof 5, 79110 Freiburg, Germany; haneke@gmx.net; 2Department of Dermatology, Inselspital, University of Berne, 3010 Bern, Switzerland; 3Private Dermatology Practice Dermaticum, 79098 Freiburg, Germany; 4Centro de Dermatología Epidermis, Instituto CUF, Senhora da Hora, 8050 Matosinhos, Grande Porto, Portugal

**Keywords:** onychomycosis, predisposing factors, foot deformation, toe malposition, *Hallux valgus*, *Hallux valgus interphalangeus*, *Hallux erectus*

## Abstract

***Introduction*:** It has long been accepted that trauma is one of the most important and frequent predisposing factors for onychomycoses. However, the role of direct trauma in the pathogenesis of fungal nail infections has only recently been elucidated in a series of 32 cases of post-traumatic single-digit onychomycosis. The importance of repeated trauma due to foot and toe abnormalities was rarely investigated. ***Aim***
*of the study*: This is a multicenter single-author observational study over a period of 6 years performed at specialized nail clinics in three countries. All patient photographs taken by the author during this period were screened for toenail alterations, and all toe onychomycosis cases were checked for whether they contained enough information to evaluate potential foot and toe abnormalities. Particular attention was paid to the presence of *hallux valgus*, *hallux valgus interphalangeus*, *hallux erectus*, inward rotation of the big toe, and outward rotation of the little toe, as well as splay foot. Only cases with unequivocal proof of fungal nail infection by either histopathology, mycologic culture, or polymerase chain reaction (PCR) were accepted. ***Results*:** Of 1653 cases, 185 were onychomycoses, proven by mycologic culture, PCR, or histopathology. Of these, 179 involved at least one big toenail, and 6 affected one or more lesser toenails. Three patients consulted us for another toenail disease, and onychomycosis was diagnosed as a second disease. Eight patients had a pronounced *tinea pedum*. Relatively few patients had a normal big toe position (*n* = 9). Most of the cases had a mild to marked *hallux valgus* (HV) (105) and a *hallux valgus interphalangeus* (HVI) (143), while *hallux erectus* was observed in 43 patients, and the combination of HV and HVI was observed 83 times. ***Discussion*:** The very high percentage of foot and toe deformations was surprising. It may be hypothesized that this is not only a pathogenetically important factor but may also play an important role in the localization of the fungal infection, as no marked *hallux* deviation was noted in onychomycoses that affected the lesser toes only. As the management of onychomycoses is a complex procedure involving the exact diagnosis with a determination of the pathogenic fungus, the nail growth rate, the type of onychomycosis, its duration, and predisposing factors, anomalies of the toe position may be important. Among the most commonly mentioned predisposing factors are peripheral circulatory insufficiency, venous stasis, peripheral neuropathy, immune deficiency, and iatrogenic immunosuppression, whereas foot problems are not given enough attention. Unfortunately, many of these predisposing and aggravating factors are difficult to treat or correct. Generally, when explaining the treatment of onychomycoses to patients, the importance of these orthopedic alterations is not or only insufficiently discussed. In view of the problems encountered with the treatment of toenail mycoses, this attitude should be changed in order to make the patient understand why there is such a low cure rate despite excellent minimal inhibitory drug concentrations in the laboratory.

## 1. Introduction

Onychomycoses are said to be the most frequent nail diseases. Having been a rare event some 150 years ago, they are now seen in 5–20% of the population depending on the age and gender of the patients, environmental factors and climate, profession, hygiene conditions, comorbidities, and habit of wearing almost airtight shoes, to mention just a few [1,2,3,4,5,6,7]. Dermatophytes, in particular *Trichophyton rubrum* and *T. mentagrophytes*, as well as new emergent nondermatophytes such as *Fusarium species*, are the most common causal agents. In the last 50 years, a wealth of information and knowledge on fungal nail infections has been accumulated; however, there is still a long way to understand all problems [8]. One of the most important risk factors is hereditary susceptibility, as has been proven in many family trees demonstrating a vertical spread of onychomycoses in the family [9,10,11]. Another question is whether onychomycoses are indeed the most frequent nail conditions, as nail changes due to skeletal abnormalities have not been considered by most authors, and these nail alterations often look very similar to toenail mycoses [12]. They are also said to be secondarily colonized by nondermatophyte fungi, which is not accepted as true onychomycosis by some authors [13].

To find out the potential role of foot and toe anomalies for fungal nail infections, the photo archive of the author was searched for toenail diseases, and all proven onychomycoses were listed, as well as the position of the toes and visible foot alterations.

## 2. Methods

This is a single-author multicenter observational study conducted in specialized nail clinics between January 2018 and December 2023 (with a three-month break during the beginning of the COVID-19 pandemic). In this 6-year period, 1665 patients with toenail disease were seen. All patients were photographed with a smartphone, and patient details (name, birth date, date of clinical examination), clinical diagnosis (diagnoses), and some other relevant findings were listed (Table 1). The photos were checked for the position of the big toe in relation to the first metatarsal bone and the potential deviation of the long axis of the distal phalanx from that of the proximal phalanx. Although radiographs were available for some toenail conditions, this was not the rule for the diagnosis of onychomycoses. Only cases with sufficient information permitting the toe and foot to be evaluated for anomalies were included in this observation.

A hallux valgus was diagnosed when there was a deviation of the axis of the first metatarsal and proximal phalanx bones of more than 3° and a hallux valgus interphalangeus when the deviation was 3° or more. A hallux erectus was diagnosed when the tip of the toe did not touch the ground in a relaxed position or the extensor tendon was visibly under tension. The inward rotation of the big toe, also called tilting, was again diagnosed with a deviation of the rotation axis of >3°. An outward rotation of the little toe of more than 15° was recognized as pathological. The measurements of the deviation angles were made with the rotation function in Microsoft 2024 Word layout.

## 3. Results

Out of 1663 patients, the axis of the toes in relation to the metatarsal bones could not be evaluated in 11 cases, thus leaving 1652 cases for this observation. Of these, 185 patients (11.2%) had an onychomycosis proven by mycological examination (direct microscopy and culture), histopathology using PAS stain, or PCR. Of the 185 fungal toenail infections, 179 involved at least one big toenail, and 6 affected one or more lesser toenails (without big toenail). Onychomycosis was diagnosed as a chance observation in three patients who consulted us for another nail condition. Only eight patients had a pronounced *tinea pedum*, for which they asked for help; however, tinea was clinically diagnosed in another 70 cases. This had not been noticed by the patients before, and some even did not want their tinea to be treated.

A normal straight big toe position was seen in only nine patients (*n* = 9, 5%). Most of the cases had a mild to marked hallux valgus (HV) (105, 58.6%) and a hallux valgus interphalangeus (HVI) (Figure 1, Figure 2, Figure 3 and Figure 4) (143, 78.1%). Hallux erectus was observed in 43 patients (23.2%), and the combination of HV and HVI was observed 83 times (44.9%) (Figure 1, Figure 2, Figure 3, Figure 4, Figure 5 and Figure 6) (Table 2).

As the patients consulted a specialized nail clinic, this observation does not reflect the normal population.

## 4. Discussion

For approximately 50 years, onychomycoses have been considered to be the most common nail diseases. This was questioned about 10 years ago, when it was claimed that nail changes due to orthopedic foot and toe anomalies would be the most common cause of toenail alterations [12,13]. This is underlined by our experience that more than 90% of all patients with toenail dystrophy showed some type of toe or foot anomaly [14,15].

It has long been known that there are predisposing factors for the development of onychomycoses [16,17]. Many of these factors are not amenable to treatment, such as male gender, advanced age, genetic susceptibility, and diabetes mellitus. Fungal nail infection is more frequent in psoriasis patients. Many surveys have shown a higher prevalence of onychomycoses in men [18], although there are considerable differences. This study also showed a male preponderance. The prevalence of onychomycoses increases proportionally with age until about 80 years [19,20].

One of the most important predisposing factors is certainly the autosomal dominant susceptibility to fungal nail infections [10]. Particularly in young patients, a family history should always be performed [9,20]. This often shows that there is a vertical spread of the infection in the family from grandparents to parents to children and grandchildren, whereas usually the spouse coming from another family remains onychomycosis-free despite year- or even decade-long contact with the infected family, provided the spouse comes from a family without this genetic susceptibility. However, household spread may also be important [11].

Sex is a potential risk factor for *tinea pedum* and onychomycosis, with considerably more males being affected by onychomycoses. The reason is not entirely clear; however, it was often assumed that men are more prone to traumatize their toes and thus render their nails more susceptible to fungal infection [21,22]. Damaged nails are more susceptible to fungal infection, particularly with *Fusarium* spp. (Figure 7 and Figure 8) [23,24].

Sports activities have been observed to enhance the risk of fungal nail infection. This may be due to increased foot and toe trauma, nonphysiologic stress and strain, suboptimal hygienic conditions, often tight special footgear, sweating, use of communal showering facilities, and many more causes [25,26,27,28]. Onychomycosis was found in 60.7% of professional football (soccer) players compared with 3.3% in a control group (Figure 9) [29].

Impaired blood supply is associated with loco-regional malperfusion, enhancing the likelihood of a fungal infection and diminishing the chances of successful antifungal therapy [30].

Chronic venous insufficiency is also associated with a higher risk of toe onychomycosis; however, nail deformations similar to fungal infections are frequent in chronic venous insufficiency. These onychomycoses have an even lower cure rate [31].

Peripheral neuropathies render the individual more prone to sustain unnoticed trauma, and trauma is an important predisposing factor [32].

Diabetes mellitus is more often associated with fungal infections of the feet and nails as it combines peripheral neuropathy with vascular insufficiency and decreased immunity [16,33,34,35,36,37,38,39]. Advanced age, male sex, diabetes, diabetic peripheral neuropathy, and lower limb ischemia are independent risk factors for developing onychomycosis [17]. The severity of diabetes mellitus is also an important risk factor. Diabetic foot syndrome, as a particularly severe manifestation of diabetes mellitus, has a very high risk of fungal toenail infection [40].

Hemodialysis patients often suffer from fungal nail infections. This is probably due to the primary cause of renal failure, like diabetic nephropathy, but hemodialysis as such may also predispose the patients to fungal nail infections [41,42,43,44].

All kinds of immune defects represent a high risk of developing an onychomycosis. This is particularly evident in transplant patients and those infected with human immunodeficiency virus (HIV) [42,45,46,47,48]. Superficial white and proximal white subungual onychomycosis are particularly suggestive of HIV infection [49].

Iatrogenic immune depression is a risk for onychomycoses [50,51]. Modern anti-inflammatory treatments with glucocorticosteroids, immunosuppressive cytostatic drugs, and many biologics render the subjects more susceptible to fungal infections, including of the nails [51,52,53,54,55,56].

Foot and toe anomalies are factors that have long been neglected (Figure 10, Figure 11 and Figure 12). The human foot is a very complex structure, consisting of 26 bones plus 2 sesamoid bones, 40 joints with 12 extrinsic and 19 intrinsic muscles, many ligaments and tendons, skin of different anatomy on the soles and dorsa, highly specialized subcutaneous tissue acting as a cushion during walking, running, and jumping, many blood vessels and specialized vascular structures, and a complicated nervous supply [57]. The two main functions of the feet are to act as a flexible support to the weight-bearing lower limbs and as a rigid lever to aid propulsion during locomotion [14,15,57]. The toes are an extension of the soles, and both increase the stability of stance and balance, as well as augment the lever action during propulsion. In the last phase of the gait, called toe-off, the entire body weight is on the tip of the big toe; however, the kinetic energy of the forward thrust increases the weight by a factor of 2.5 during normal walking speed, and this is even further increased with many sports activities where running and acceleration are important. The nails, particularly the big toenail, give counterpressure to the forces acting on the soft tissue of the toe tip and prevent it from being dislocated dorsally and forming a distal bulge [14]. For this important function, the nail is anatomically linked to the tendons and ligaments of the distal interphalangeal joint to one functional unit. This is also why the nail has been called a musculoskeletal appendage [58,59]. The nail consists of the matrix that produces the nail plate, and the nail bed that attaches the nail firmly to the underlying nail bed dermis and bone, as well as the periungual structures, such as the proximal and lateral nail folds. Although being functionally very different, the nail bed and matrix must work perfectly together in order to have a normal nail. If one of these structures is at fault, there will be no normal nail; this is evidenced in the condition called the disappeared nail bed [15]. The cuticle protects the nail pocket, also called the cul-de-sac, from the penetration of foreign substances and microbes. The hyponychium has a similar function, anchoring the nail plate to the distal end of the nail bed and preventing the penetration of foreign bodies under the nail [60]. It is probably also a barrier to fungal infection of the distal nail bed [57].

Malposition of the nail, be it due to malalignment of the nail itself, displacement of the distal phalanx, or hallux valgus, results in chronic repeated microtrauma to the nail, resulting in subungual hyperkeratosis, onycholysis, paronychia, Beau’s lines, onychomadesis, and onychomycosis [57,61,62]. Onycholysis, too short a nail, or lack of the nail of the big toe after nail avulsion or traumatic nail loss, leads to a distal bulge and a shortened or disappeared nail bed [63,64]. The bulge is part of the toe tip and covered with ridged skin; there is no epidermization of the nail bed, as is easily seen by the sharp delimitation between the bulge and the shortened nail bed [65].

As briefly mentioned above, nail alterations caused by orthopedic abnormalities may look very much like fungal nail infections, particularly distal–lateral subungual onychomycosis [12,13,66,67]. This observational real-world study has shown that “normal” straight toes are rather the exception than the rule. Approximately 90% of the onychomycosis patients had a toe malposition, mostly a hallux valgus interphalangeus (HVI) or hallux valgus (HV), and often both, whereas hallux erectus (HE, hyperextension) was less frequent. Splay foot is commonly associated with flat foot (although not evaluable in this study due to the type of clinical photography performed), and they in turn often lead to the inward rotation of the big toe and outward rotation of the little toe [57]. These orthopedic conditions lead to nonphysiologic strain on the nails by direct repeated trauma or by the modification of pre-existing nail alterations. Friction may cause subungual hyperkeratosis that breaches the integrity of the hyponychium and allows fungi to penetrate through it and reach the nail bed; in extreme cases, it may cause a subungual corn (heloma subungual). However, it has to be stressed that subungual hyperkeratosis and true nail thickening are often erroneously seen as the same [67]. A malaligned nail or a laterally displaced toe is prone to sustain trauma through torsion stress forces during walking, crawling, or from footwear [62]. It appears that compression forces that hit a deviated nail are of particularly devastating action on the attachment of the nail to the nail bed [15]. However, we have very rarely found pathogenic fungi in congenital malalignment of the big toenail [14,15].

Patients consulting a specialized nail clinic for long-standing onychomycosis have usually seen several physicians, including dermatologists before. They are embarrassed by their infection and often have low self-esteem, decreased quality of life, not-infrequent pain, abstain from social contacts, and most have already undergone several treatments [68]. They are challenging patients as unsuccessful therapies have left them frustrated and even angry that they paid for their medication “for nothing” but with the risk of potentially serious adverse effects. It is of paramount importance to explain the problems of any onychomycosis management and that this is not only a nail issue but a problem of the whole organism. Treatment failure of fungal toenail infections is often not because of insufficient drug activity, whether applied topically or given systemically, but a failure to recognize the many risk factors discussed above [57]. However, even in the podiatric literature, foot, toe, and gait anomalies in onychomycosis patients are often ignored [69]. Chronically traumatized nails tend to develop a massive subungual hyperkeratosis. It was found that their thickness is inversely related to treatment success [70].

Yet another unsolved problem is the frequent occurrence of recurrences. In most cases, it is not clear whether this is due to a relapse of residual disease or a true infection [71]. It is known that spores, also of dermatophytes, remain viable for years. Fungus-containing little scales are lost where individuals with a *tinea pedum* walk barefoot, particularly in their own bathrooms and bedrooms. Stepping on such keratin flakes makes them stick on the sole of the foot, where they can form invasive hyphae within about 4 h. The role of dermatophytoma and similar masses of fungi in the nail bed compressed between the living epithelium and the overlying nail has not yet been elucidated; the fungi here may have cell walls up to 20 times thicker than in common hyphae, rendering them intrinsically more resistant, and they are even more difficult to reach by both topical and systemic antifungal drugs (Figure 13 and Figure 14) [72,73]. Histopathology shows both dermatophytomas (Figure 15) as well as biofilms (Figure 16); these biofilms are more often bacterial than fungal and extremely common at the undersurface of onycholytic nail plates. It is known that biofilms are up to 1000 times less sensitive to antibiotics than the same microbes as single organisms [74]. Further, histopathology can prove that *Fusarium* spp. and other nondermatophyte molds are able to produce real onychomycoses (Figure 17).

The differential diagnosis is yet another problem, particularly in general practice. It is strongly recommended to have the diagnosis confirmed before starting treatment, particularly with systemic drugs [67,75,76,77].

## 5. Conclusions

Onychomycoses are said to be the most frequent nail diseases, making up approximately half of all ungual diseases. Of all dermatomycoses, they are the most difficult to treat. This is the consequence of the particular anatomy of the nail infection, the difficulty of the drugs to reach the really infected part, and the innumerable factors accompanying the fungal nail infection, of which many are not amenable to therapy or difficult to treat and are often beyond the scope of the dermatologist. Thus, onychomycosis therapy requires a well-informed and understanding patient [15,57].

### 5.1. Limitations of the Study

This is a retrospective single-examiner real-world study from different European cities with no control group. There is probably a high degree of bias, as the patients referred to the highly specialized nail clinic were almost all pre-examined and pretreated and therefore do not reflect the average. No statements can be made about the frequency of onychomycoses or foot and toe malformations in the general population.

### 5.2. Strengths of the Study

This is the largest study on onychomycoses in foot and toe anomalies to date. The level of expertise was always the same.

## Figures and Tables

**Figure 1 jof-10-00399-f001:**
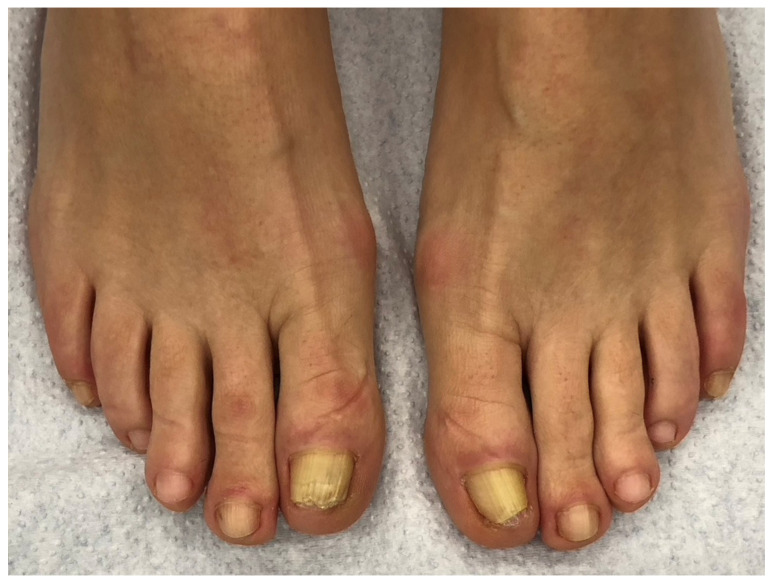
Onychomycosis of both big toes in a 49-year-old patient with mild *hallux valgus* and *hallux valgus interphalangeus*; the second toes are longer than the first ones (Greek foot). Mycologic culture revealed *Trichophyton rubrum*. Note that both big toenails are thick and yellowish, with marked subungual hyperkeratosis, and the *extensor hallucis longus* tendon is visibly taut.

**Figure 2 jof-10-00399-f002:**
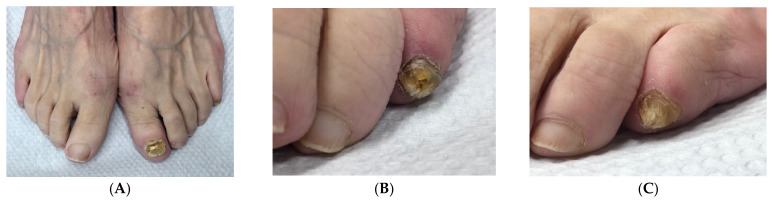
A 63-year-old woman with dystrophic onychomycosis of the left big toenail and both little toes and marked *hallux valgus*: (**A**) Overview. (**B**) Right little toenail. (**C**) Left little toenail.

**Figure 3 jof-10-00399-f003:**
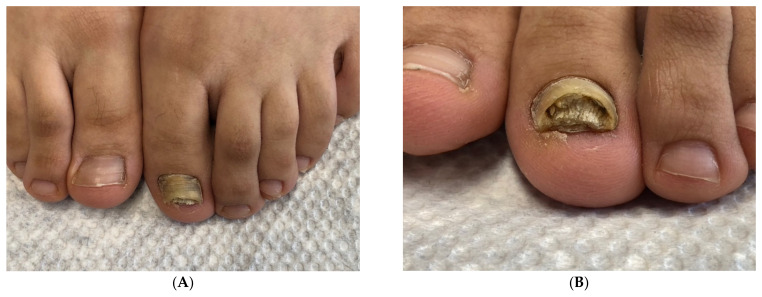
Hypertrophic onychomycosis of the left big toe in a 21-year-old woman with mild *hallux valgus* and *hallux valgus interphalangeus*. Note the desquamation at the hyponychium and toe tip, indicative of tinea pedis: (**A**) Overview. (**B**) Close-up.

**Figure 4 jof-10-00399-f004:**
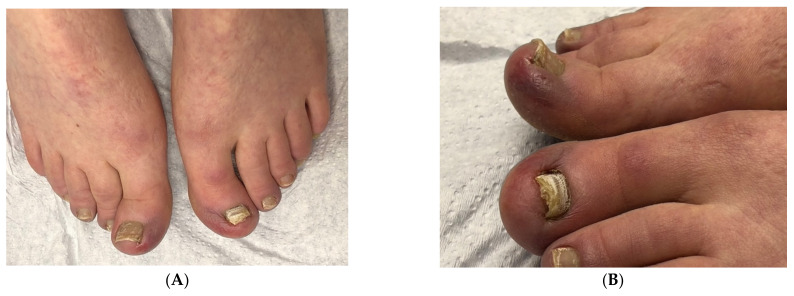
Onychomycosis in a 30-year-old woman with *hallux valgus, hallux valgus interphalangeus* and *hallux erectus*: (**A**) Dorsal view. (**B**) Side view.

**Figure 5 jof-10-00399-f005:**
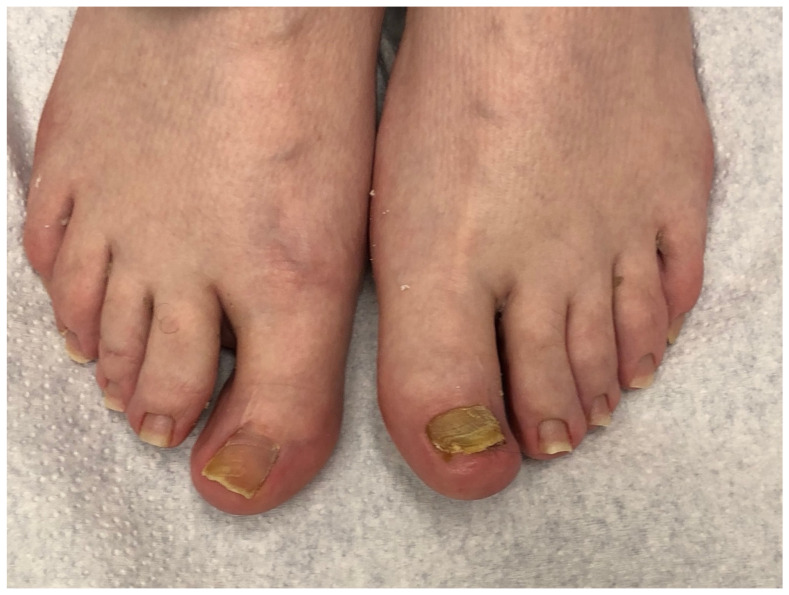
Single-digit posttraumatic onychomycosis in a 55-year-old man with *hallux valgus*, *hallux valgus interphalangeus*, left *hallux erectus*, short left big toenail, and markedly shrunken nail bed.

**Figure 6 jof-10-00399-f006:**
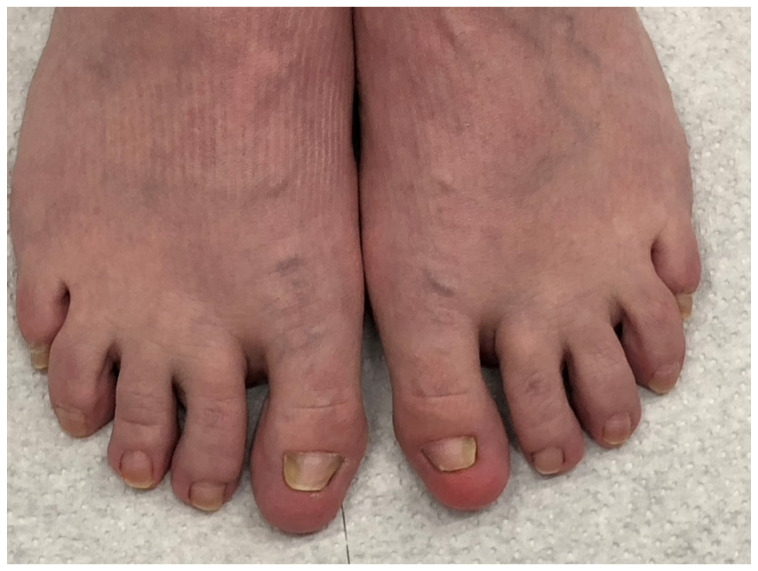
Big toes of a 51-year-old man with longstanding onychomycosis of both big toes, *hallux valgus interphalangeus* and mild *hallux valgus*, inward rotation of the big toes (left more than right), short nails, and a large distal wall covered with ridged skin of the tip of the toe. The patient had always cut his nails as short as he could.

**Figure 7 jof-10-00399-f007:**
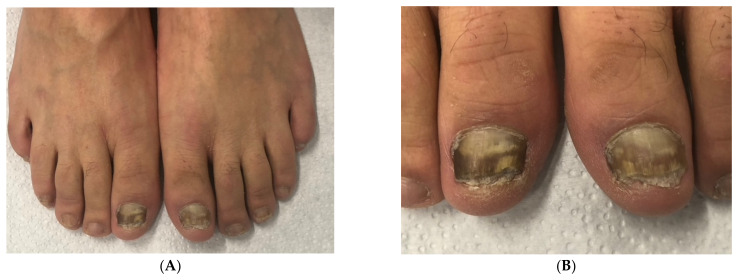
Big toes of a 61-year-old previous professional soccer player with onychomycosis of both big toes. Note that there is only a mild deviation of the distal phalanx, hinting at the importance of previous professional soccer playing as a predisposing factor: (**A**) Overview. (**B**) Close-up photograph.

**Figure 8 jof-10-00399-f008:**
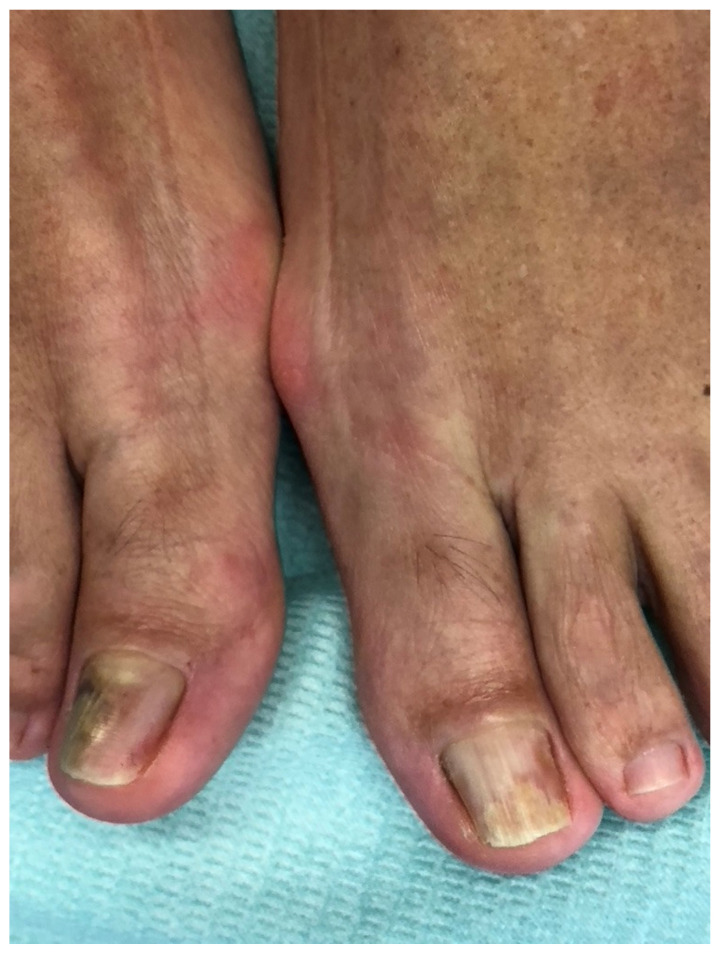
Onychomycosis of the big toes due to *Fusarium solani* complex. There is a marked *hallux valgus* of the left big toe and a pronounced *hallux valgus interphalangeus* of the right big toe.

**Figure 9 jof-10-00399-f009:**
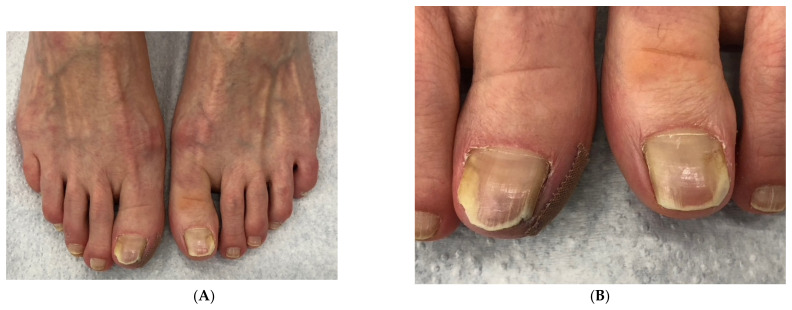
*Fusarium oxysporon* onychomycosis in a 53-year-old woman with mild *hallux valgus* and *hallux valgus interphalangeus*: (**A**) Overview showing the taut *extensor hallucis longus* tendons. (**B**) Close-up.

**Figure 10 jof-10-00399-f010:**
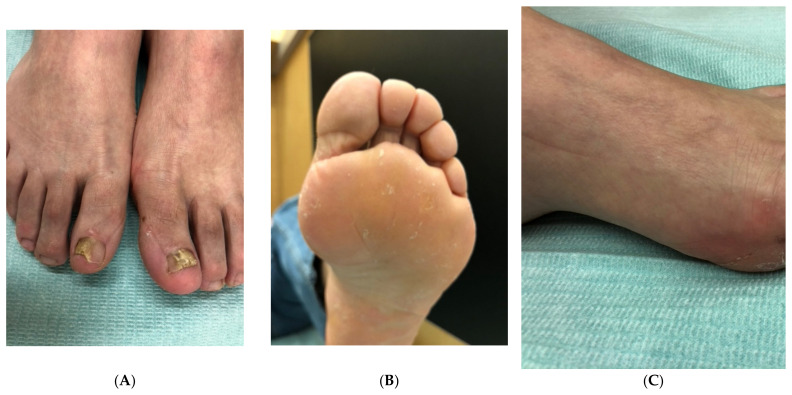
A 45-year-old woman with *hallux valgus*, onychomycosis, and *tinea pedum*: (**A**) Dorsal view. (**B**) Plantar view. (**C**) Medial view of the left foot showing a *hallux erectus*.

**Figure 11 jof-10-00399-f011:**
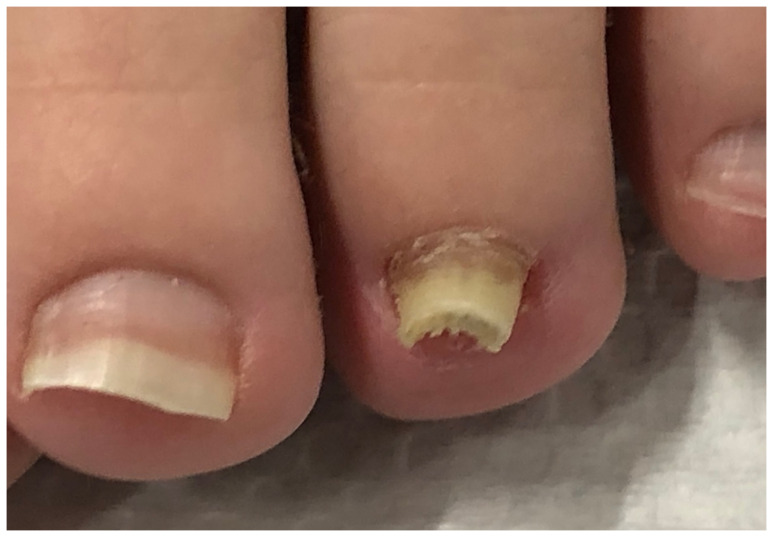
Isolated onychomycosis of the left 3rd toe in a 27-year-old woman.

**Figure 12 jof-10-00399-f012:**
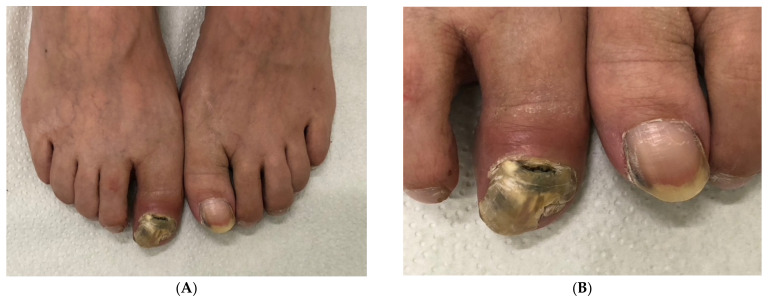
Post-traumatic onychomycosis in a 59-year-old man with *hallux valgus interphalangeus*: (**A**) Overview. (**B**) Close-up photograph.

**Figure 13 jof-10-00399-f013:**
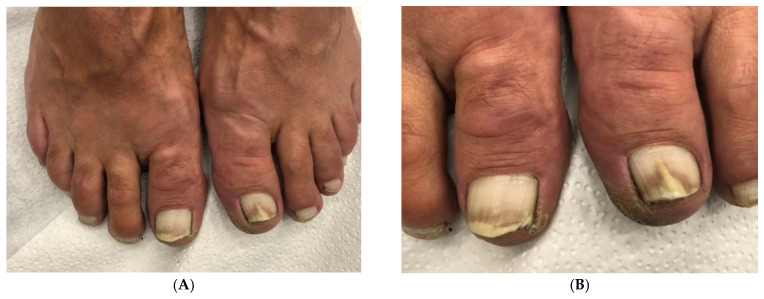
Dermatophytoma of the left great toenail in a 48-year-old man with diffuse leukonychia: (**A**) Overview. (**B**) Close-up.

**Figure 14 jof-10-00399-f014:**
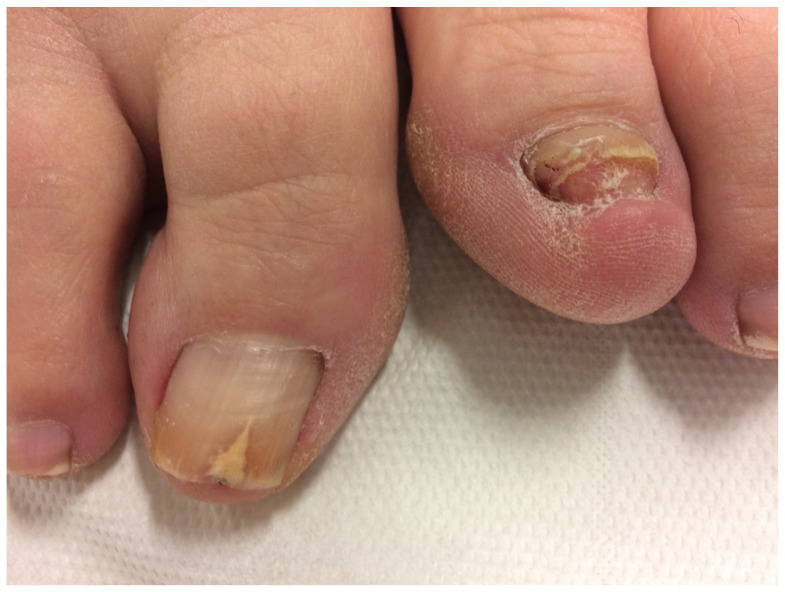
A 53-year-old female patient with right *hallux valgus interphalangeus* and dermatophytoma, as well as left big toe with *hallux erectus*, disappeared nail bed (shrunken nail bed), and big distal bulge.

**Figure 15 jof-10-00399-f015:**
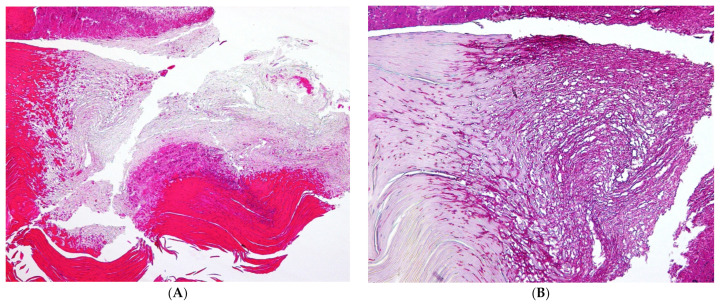
Dermatophytoma: (**A**) Hematoxylin and eosin stain: the keratin is red, part of the fungal masses is reddish violet, and the pale gray structures are fungal elements, which are normally not stained by H&E. Original magnification 100×. (**B**) PAS stain: the fungi are stained intensely violet; original magnification 200×.

**Figure 16 jof-10-00399-f016:**
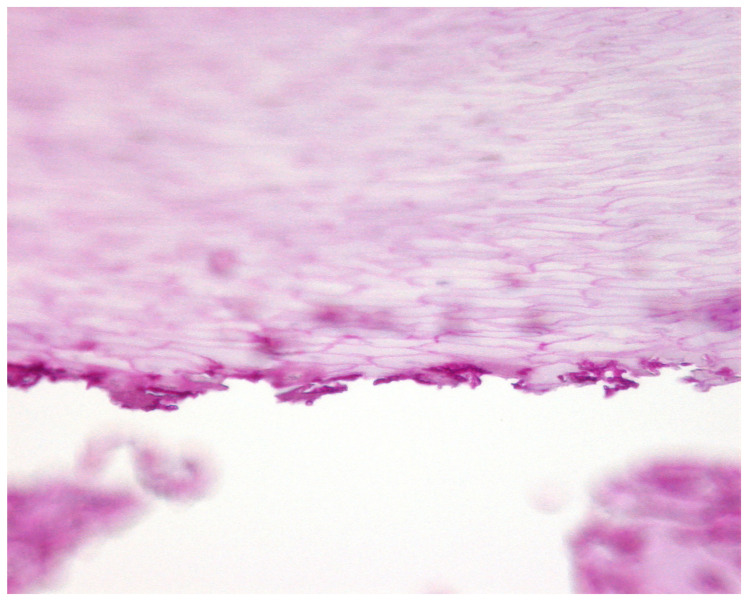
Biofilm of *Fusarium solani* at the undersurface of the nail plate. PAS stain, original magnification 400×. Fungal biofilm at the undersurface of the nail plate.

**Figure 17 jof-10-00399-f017:**
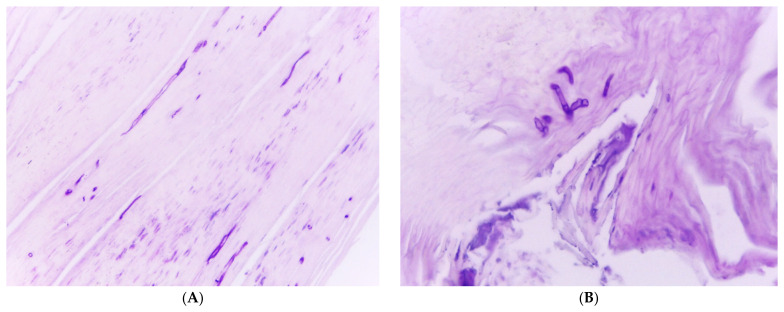
Onychomycosis due to *Fusarium solani* in a 32-year-old woman: (**A**) Fungal filaments of variable diameter with occasional septae in the nail, which also contains remnants of neutrophils: (**B**) Branching filaments. Histopathology, PAS stain, original magnification 400×.

**Table 1 jof-10-00399-t001:** Demographic data.

Age [Years]	Gender	
	Female 84 (45.4%)	Male 101 * (54.6%)
0–9	1	1
10–19	4	2
20–29	9	10
30–39	11	5
40–49	22	23
50–59	19	24
60–69	13	20
70–79	5	14
>80	0	1

* The age could not be retrieved in one male patient.

**Table 2 jof-10-00399-t002:** Podological results of the onychomycosis patients.

	Number	Percent
Normal straight toe	9	5
Hallux valgus (HV)	105	58.6
Hallux valgus interphalangeus (HVI)	143	78.1
HV plus HVI	83	44.9
Hallux erectus (HE)	43	23.2
HV + HVI + HE	25	10.8
HVI + HE	6	3.2

## Data Availability

The original contributions presented in the study are included in the article, further inquiries can be directed to the author.

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
