# Peer review of "Onychomycosis in Foot and Toe Malformations"

_jof, 2024, doi:10.3390/jof10060399_

Round 1

Reviewer 1 Report

The manuscript described  the relation the consequence of the particular anatomy of the nail infection

The manuscrit presents many figures. Please choose some figures to illustrate the problem ( only 3).   

Author Response

Thank you very much for your review and comment.

The other reviewers did not want me to reduce the number of figures so I left them in the manuscript; however, I do not mind to skip 10 figures if the editor thinks there are too many figures.

Reviewer 2 Report

The manuscript focuses on a very important topic such as onychomycoses the most frequent nail diseases. The study aims to show trauma as one of the most frequent factors for onychomycoses. It is a multicenter single-author observational 6-year study (2018-2023) conducted on nail clinics in three European countries including 185 patients with proven onychomycoses. It was observed a very high percentage (95%) of foot and toe deformations in this group. Additionally, the author revises other predisposing factors for the development of onychomycoses and common problems linked to the frequent occurrence of infection recurrences. The study stresses that onychomycosis therapy requires a well-informed and understanding patient.

The 17-photo archive of the author presented is very didactic and illustrative for dermatologists and laboratory technicians.

onychomycoses the most frequent nail diseases. The study aims to show trauma as one of the most frequent factors for onychomycoses. 

It is a original multicenter single-author observational 6-year study (2018-2023) conducted on nail clinics in three European countries including 185 patients with proven onychomycoses. It was observed a very high percentage (95%) of foot and toe deformations in this group. Additionally, the author revises other predisposing factors for the development of onychomycoses and common problems linked to the frequent occurrence of infection recurrences. 

The study covers the gap that onychomycosis therapy requires a well-informed and understanding patient and not only adherence to treatment. Moreover, there are many factors that are beyond the dermatologist assistance.

The 17-photo archive of the author presented is very didactic and illustrative for dermatologists and laboratory technicians. 

The study shows the relevance of trauma and deformities as one of the most frequent factors for onychomycoses. 

The methodology is adequate. It could be useful to give some information on the frequency of foot fingers deformity in normal population, showing no onychomycoses lesions. 

The conclusions are consistent with the data provided by the observational study.

The references are appropriated.

Line 13. Use plural for specialized nail clinics. Revise italics for Latin names

Author Response

Thank you very much for your positive review and comments.

"The methodology is adequate. It could be useful to give some information on the frequency of foot fingers deformity in normal population, showing no onychomycoses lesions." There are two problems with giving informations on the frequency of toe deformities in the normal population: 1. The patients described here are "selected" patients as seen in a specialized nail clinic after having been seen and treated in other dermatological practices. 2. The patients are not representing the "normal" population. Thus, these patients cannot be comnpared with the normal population. However, the presence of "normal (fore)feet and toes" in patients with other toenail affections was also only about 10%. Thus, what we can deduct from these studies is that toe and foot deformations render patients more susceptible to toenail changes.

"Line 13. Use plural for specialized nail clinics." Thank you for your comment. I apologize for having overlooked this grammatical error.

"Revise italics for Latin names." All Latin scientific names (except those that are commonplace even in normal laare now italicized which is highlighted in yellow.

Reviewer 3 Report

The manuscript “Onychomycosis in foot and toe malformations” is a multicenter observational study carried out over six years in a specialized nail clinic and aimed to understand the possible role of anomalies of the feet and toes in infections due to nail fungus. It is an interesting, well-structured article, considering that onychomycosis is widely distributed worldwide and the number of cases is high. However, I have some comments.

I consider that the author should mention in the introduction the causal agents of onychomycosis and mention the epidemiological aspects since the majority of onychomycosis is caused by dermatophytes (Trichophyton rubrum and T. mentagrophytes among the most common organisms); however, the worldwide prevalence of onychomycosis caused by non-dermatophyte fungi has increased in recent years (yeasts, Scopulariopsis brevicaulis, Fusarium spp., Aspergillus spp., Scytalidium dimidiatum and Acremonium spp. among the common organisms).

The author must follow the order of the template provided by the journal; therefore, he must correct the order of the different sections of the manuscript.

Ensure that the entire manuscript has the same type of font suggested by the magazine (Palatino Linotype).

Review the manuscript and correct scientific names; they should be italicized.

Tables 1 and 2 should be placed after the “Results” section.

In table 1, place “Gender” in the correct place.

Although the figures are a fundamental element of the work, I suggest that they be adapted so that on each page, a single figure is placed (A, B, or C, if applicable), including the legend of the figure, and placed after the tables, before of the “Discussion” section.

References must be carefully reviewed and strictly follow the journal format.

Author Response

Thank you very much for your valuable comments.

  1. The most important causal agents of onychomycoses, such as Trichophyton rubrum, T mentagrophytes as well as the emergent Fusarium spp are mentioned in the introduction. The other  non-dermatophytes are not mentioned as they were not observed in this study.
  2. The font Palatino Linotype is used throughout the entire manuscript.
  3. All efforts were made to arrange figures to one page, after the tables and before the discussion. However, this is usually done by the journal editor.
  4. All references are carefully checked. 
  5. "The author must follow the order of the template provided by the journal; therefore, he must correct the order of the different sections of the manuscript." The different sections of the manuscript are now ordered as follows: abstract, keywords, acknowledgements and statements, introduction, methods, results, tables, discussion, conclusion, limitations, strength of the study, and references. Following are the figure legends and at last the figures with their captions.
  6. "Tables 1 and 2 should be placed after the “Results” section." The tables are now arranged as suggested. "Gender" is placed correctly.
  7. "Although the figures are a fundamental element of the work, I suggest that they be adapted so that on each page, a single figure is placed (A, B, or C, if applicable), including the legend of the figure, and placed after the tables, before of the “Discussion” section." The figures are now adapted so that they are placed on one page. I would like to leave it to the journal editor to place the figures according to space.
  8. "References must be carefully reviewed and strictly follow the journal format." The reference style was compared with another article in J Fungi and are now in journal style.
  9. All changes, corrections and improvement of the manuscript are highlighted in yellow.